# Traditional Medicine Plant, *Onopordum acanthium* L. (Asteraceae): Chemical Composition and Pharmacological Research

**DOI:** 10.3390/plants8020040

**Published:** 2019-02-12

**Authors:** Ekaterina Robertovna Garsiya, Dmitryi Alexeevich Konovalov, Arnold Alexeevich Shamilov, Margarita Petrovna Glushko, Kulpan Kenzhebaevna Orynbasarova

**Affiliations:** 1Department of Pharmacognosy, Botany and Technology of Phytopreparations, Pyatigorsk Medical-Pharmaceutical Institute (PMPI), Branch of Volgograd State Medical University, Ministry of Health of Russia, Pyatigorsk 357532, Russia; d.a.konovalov@pmedpharm.ru (D.A.K.); shamilovxii@yandex.ru (A.A.S.); perla21@yandex.ru (M.P.G.); 2South Kazakhstan State Pharmaceutical Academy, Shymkent 160019, Kazakhstan; kulpan_ok@mail.ru

**Keywords:** *Onopordum acanthium*, Asteraceae, metabolites, pharmacology, antitumor, anti-inflammatory activity

## Abstract

For many years, plants have been used in the traditional medicine of different cultures. The biennial plant of the family Asteraceae, *Onopordum acanthium* L., also known as Scotch thistle, is used in traditional medicine as an anti-inflammatory, antitumor, and cardiotonic agent. The plant is widespread in the world; it grows in Europe and Asia and was introduced to America and Australia. Stems and buds of the first-year plant are used in cooking as an analogue of artichoke in European cuisine. Additionally, inflorescences contain a complex of proteolytic enzymes “onopordosin”, which may be used as a milk-clotting agent in the dairy industry. The chemical composition of the aerial part and roots of *O. acanthium* is represented by flavonoids, phenylpropanoids, lignans, triterpenoids, sesquiterpene lactones, and sterols. The anti-inflammatory, antiproliferative, and cardiotonic properties of the plant have been confirmed by pharmacological experiments with extracts and individual compounds using in silico, in vitro, and in vivo methods. This work is a review of information on the chemical composition and pharmacological studies of *O. acanthium* as a promising medicinal plant.

## 1. Introduction

In 1503, the Scottish poet William Dunbar wrote an allegory for the wedding of King James IV and Princess Margaret Tudor, called “the Thistleand the Rose”, where he presented James as a lion, eagle, and thistle, Margarita was presented as a rose. This allegory describes James (known as “James the just”) as the lion, the guardian of peace in England and Scotland as the eagle and protector of men as the thistle. In Scotland, the prickly thistle (*Onopordum acanthium* L., Scotch thistle, cotton thistle) is known as a national symbol, although the common thistle (*Cirsium vulgare* (Savi) Ten.) is more often depicted [1].

The genus *Onopordum* includes about 50 species. The species *O. acanthium* L. is widely distributed. According to The Plant List [2], other names of *O. acanthium* are: *Acanos spina* Scop., *Acanthium onopordon* Gueldenst, *Carduus acanthium* (L.) Baill., *Onopordum acanthium* var. *acanthium*, and *Onopordum acanthium* subsp. *acanthium*. People of different countries refer to the plant as Scotch thistle, cotton thistle, heraldic thistle (English), eselsdistel, krebsdistel (German.), and chardon aux ânes (French). In Greek, the genus name translates as *onos —* donkey, as pordon — carminative, *acanthium* species name comes from the Greek word meaning “prickly”.

According to the taxonomy of vascular plants of A. L. Takhtajan [3], *O. acanthium* belongs to the family Asteraceae, subfamily Carduoideae (Lactucoideae), tribe *Cardueae*, subtribe *Carduinae*, genus *Onopordum* L., and section *Onopordum*. The superfamily taxonomy consists of the kingdom Plantae, subkingdom Viridiplantae, infrakingdom Streptophyta, superdivision Embryophyta, division Tracheophyta, subdivision Spermatophyta, class Magnoliopsida, superorder Asteranae, and order Asterales [4]. The modern systematics of vascular plants (Angiosperm Phylogeny Group, APG) are based on the comparison of gene sequences in chloroplasts and ribosomes. Last updated in 2016, the APG IV uses hoards in which all taxa in a hierarchical sequence have only one-to-one correspondence. The family Asteraceae corresponds to the clade Asterids, order Asterales Link [5].

There are two independent subspecies of *O. acanthium*: *Onopordum acanthium* subsp. *ceretanum* (Sennen) Arènes and *Onopordum acanthium* subsp. *gautieri* (Rouy) Bonnier [2].

## 2. Methodology of Review

We used scientific databases, such as PubMed, ScienceDirect, Mendeley, ResearchGate, and Google Scholar. Research was carried out using the keywords “onopordum”, “acanthium”, “thistle”, and “onopordon”. We obtained about 600 references according to these key words. Bibliographic data were managed using the Zotero 5.0.60 software (Center for History and New Media at George Mason University, Virginia, USA). References relating to *Onopordum acanthium* L. consist of 103 English and 53 Russian sources including common references (six English and 13 Russian), patents (14 Russian), botanical characteristics (six English and three Russian), chemical and pharmacological studies (42 English and 20 Russian), ecological and biological works (35 English and two Russian), industrial studies (six English and one Russian), and reviews about traditional medicinal practice (eight English). All data were statistically analyzed in corresponding articles.

## 3. Botanical Characteristics

*O. acanthium* L. is a biennial herb that grows to 50–200 cm in height. The radix is thick and succulent above 30 cm. The stem is upright, branching, has wings less than 1.5 cm wide with spines on the edge. The stem is round and 2–3 cm in diameter. The first-year plants grow in a rosette of leaves, and there are broad, elliptic lanates on both sides. In the second year, the plants flower, bear fruits, and die. The rosette leaves are elliptic and broad and have petioles and lanates on both sides. The stem leaves are smaller, sessile, and oblong. All leaves have stout yellow spines on the edge and thick pubescence. On the bottom of stems there are one or several inflorences. Capitula are globular, have spiny involucre which contain green leaves with yellow hard spines on the ends. The receptacle is alveolate. The calyx consists of hairs. The corolla is purple and flowers are hermaphrodite. The achenes are brown and obovate and have ribs on the surface and redhead pappuses which are twice as long as the achenes [6].

*O. acanthium* blooms in June–September and achenes ripen in July–October. The plant grows on stone or sandy soil rich in ammonium salts in light open areas [7].

## 4. Ethnomedical Usage

*O. acanthium* is is used in Europe as an edible plant. Roots, shoots, and inflorescences of first-year plants are used as a substitute for artichoke [8]. In Patagonia, Argentina, thistle is pollinated by *Apis mellifera* L. and is a source of honey. The plant is also an edible culture for goats and sheep [7].

In folk medicine, *O. acanthium* is used for the treatment of different types of cancer. Additionally, the powder, juice, and decoction of the aerial part of the plant is known as a diuretic, to treat nervousness. Furthermore, *O. acanthium* may stimulate the central nervous system and has cardiotonic and haemostatic properties [9]. Moreover, the infusion of leaves and inflorescences decreases edema of various etiologies [10].

*O. acanthium* and *O. illyricum* are noted in De Materia Medica in Volume III, “roots, semina and herbs”, by Berendes [11]. The fresh herb of “Cardvi tomentosi” was first mentioned in the Russian Pharmacopeia (Pharmacopoea Rossica, 1798) [12]. In the Encyclopedia of Traditional Chinese Medicines [13], *O. acanthium* is described as a source of onopordopicrin and eriodictyol. Also, the aerial part of the plant is used as a hemostatic agent.

In homeopathy, *O. acanthium* is a component of cardiotonic remedy. Flowers of *O. acanthium* and *Primula veris* L. and leaves of *Hyoscyamus niger* L. are used in Primula comp. (WALA-R, subsidiary of WALA Heilmittel GmbH, Moscow, Russia) in granules. Another composition is Cardiodoron^®^ Trophen. There are drops with ethanol extracts of fresh flowers of *O. acanthium* and *P. veris* and fresh herb of *H. niger*. These drugs are used in cardiovascular diseases and as sedative agents [14,15,16,17,18]. Gattefossé (Saint-Priest, France) has developed a cosmetic preparation containing alcohol extracts from flowers, leaves, and stems *O. acanthium.* This product may be used in creams and ointments and has moisturizing and anti-aging properties due to epidermal restructuring activity [19].

## 5. Chemical Composition

Compounds of phenol, triterpene, and steroid structures are detected in the aerial part of *O. acanthium*. Additionally, the composition of fat oil has been studied in the seeds. Sesquiterpene lactones and polyacetylenes are found in the roots.

### 5.1. Phenols

An inhibitor of angiotensin-converting enzyme (ACE) was isolated from methanol extract of achenes in an amount of 70 mg from 100 g achenes. It is 1, (E)-1-oxo-3,4-dihydro-1 H-isochromen-7-yl 3-(3,4-dihydroxyphenyl) acrylate (Figure 1) [20].

Kos [21] investigated the phenol and flavonoid contents of methanol, ethanol, and acetone extracts of flowers and leaves of *O. acanthium* L., *Carduus acanthoides* L., *Cirsium arvense* (L) Scop., and *Centaurea solstitialis* L. from the family Asteraceae. The content of phenols in flowers of *O. acanthium* was 19.71 mg of gallic acid/L of ethanol extract, 24.70 mg of gallic acid/L of methanol extract, 13.94 mg of gallic acid/L of acetone extract. The content of phenols in the leaves of *O. acanthium* was 26.34 mg of gallic acid/L of ethanol extract, 30.47 mg of gallic acid/L of methanol extract, and 36.67 mg of gallic acid/L of acetone extract.

Habibatni [22] also determined the content of phenols in butanol extract of *O. acanthium* leaves (8.96 mg of gallic acid/100 mg of dry extract).

#### 5.1.1. Flavonoids

Flavones, flavonols, and flavanones aglycons have been described in *O. acanthium*. Glycoside forms of apigenin, quercetin, and luteolin have been discovered in the aerial part of the plant (Table 1) [23].

Apigenin and luteolin were isolated from the aerial part of *O. acanthium* in the amount of 4.5 mg per 4.4 kg of dry raw materials. They are also found in the leaves and flowers [22,23,24,25].

4’-methyl ether of scutellarein (6-hydroxy apigenin) was found in leaves in the amount of 9.5 mg per 215 g of dry leaves [26]. Other methoxy derivatives (nepetin, chrysoeriol, hispidulin, pectolinarigenin) were detected in the aerial part in the amount of 8.0 mg/4.4 kg dry raw material (nepetin), 3.5 mg/4.4 kg dry raw material (hispidulin), as well as in leaves and 1.6 mg/215 g dry weight (pectolinarigenin) and flowers (chrysoeriol) [23,24,25,26].

Eriodictyol and quercetin were identified in flowers [23]. Apigenin, luteolin, and quercetin glycosides were found in herb, while anthocyanin cyanin was found in flowers (Figure 2) [23,27,28].

Kos [21] measured the content of flavonoids in several extracts of flowers and leaves of *O. acanthium*. In flowers, the content was 30.37 mg of quercetin/L of ethanol extract, 42.09 mg of quercetin/L of methanol extract, and 32.40 mg of quercetin/L of acetone extract. The content of flavonoids in the leaves of *O. acanthium* was 40.06 mg of quercetin/L of ethanol extract, 53.18 mg of quercetin/L of methanol extract, and 85.37 mg of quercetin/L of acetone extract.

Habibatni [22] determined the flavonoid content in butanol extract of *O. acanthium* leaves (3.93 mg of catechin/100 mg of dry extract).

#### 5.1.2.Phenylpropanoides

Capitula accumulate caffeic and chlorogenic acids [23]. Methanol extract from achenes contains 10.2 ± 0.2% (by volume) or 3.5 ± 0.4 mg/g of fresh seeds of isochlorogenic acid [29]. Phenylpropanoid aconiside was isolated from dry achenes in chloroform extract [30] (Figure 3).

#### 5.1.3. Lignans

Lajter [24,25,31] isolated lignans such as pinoresinol (5.5 mg/4.4 kg air-dried herb), syringaresinol (9.4 mg/4.4 kg air-dried herb) and medioresinol (3.4 mg/4.4 kg air-dried herb) (Table 2).

The roots of *O. acanthium* contain nitidanin diisovalerianate [24,25,31]. Daci [29] found in methanol extract of achenes 87 ± 2% (by volume) or 38.0 ± 3.2 mg/g of fresh fruits of arctiin (Figure 4).

#### 5.1.4. Coumarins

Bogs and Bogs [27] detected aesculin and aesculetin in herb of *O. acanthium* L. (Figure 5).

### 5.2. Terpenoids

Different parts of *O. acanthium* L. contain sesquiterpene lactones and triterpenoids.

Ivanova [32] detected iridoids in the herb of *O. acanthium*, including harpagide. There were 5.76% of iridoids in the *O. acanthium* herb.

#### 5.2.1. Sesquiterpene Lactones

Parts of *O. acanthium* contain derivatives of elemane, germacrane, eudesmane, and guaiane. These compounds may play roles in the antitumor activity of *O. acanthium*.

##### Guaianolides

In a review article [33] of the genus *Onopordum,* derivatives of guaiane are described: 4β,15-dihydro-3-dehydrozaluzanin C (estafiatone), zaluzanin C, and 4β,15,11β,13-tetrahydrozaluzanin C (dihydroestafiatone) in *O. laconicum*.

Csupor-Löffler [31] isolated these lactones from roots of *O. acanthium* (Table 3).

##### Germacranolides

Droźdź [34] isolated onopordopicrin from fresh leaves of *O. acanthium.* A total of 3.5 g of onopordopicrin was obtained from 10 kg of fresh leaves using water extract and purification with chloroform. Later, onopordopicrin was isolated from leaves [35]. Arctiopicrin was also found in *O. acanthium* herb (Figure 6) [27].

##### Elemanolides and Eudesmanolides

Watanabe [26] associated the widespread use of *O. acanthium* with allelopathy. As chemical agents, flavonoids such as pectolinarigenin (**1**), scutellarein 4’-methyl ether (**2**), two sesquiterpene lactones elemanolide 11(13)-dehydromelitensin β-hydroxyisobutyrate (**3**) and acanthiolide (**4**) (Figure 7) were found in leaves. Allelopathy was observed by the inhibition of wheat coleoptile intergrowth. The most active component was lactone **3** (half maximal inhibitory concentration (IC_50_) = 1.794 × 10^−4^ mg/mL), followed by pectolinarigenin (IC_50_ = 1.263 × 10^−3^ mg/mL), while scutellarein 4’-methyl ether was less active (IC_50_ = 1.709 × 10^−3^ mg/mL). Component **4** was produced in a small amount, and its activity was not investigated. The yields of compounds **3** and **4** were 2.2 mg and 0.9 mg, respectively, from 215 g of air-dried leaves.

#### 5.2.2. Triterpene Alcohols

Khalilov et al. isolated triterpene alcohols—α- and β-amyrin, lupeol, and triterpenoid taraxasterol—and their acetates from the leaves, stems, flowers, parts of capitula and achenes of *O. acanthium* (Figure 8). Air-dried plant material was crushed and then extraction by chloroform was performed in the ratio 1:4. The residue after chloroform evaporation was extracted by 70% ethanol. Polar compounds and chloroform residue were separated in a column with silica gel impregnated with silver nitrate. There were three fractions: fraction A included the amount of higher alkanes, fraction B contained a mixture of triterpene alcohols, and fraction C contained individual compounds such as taraxasterol or taraxasteryl acetate. Acetate accumulates most in reproductive plant parts. The structure of substances was defined by the methods of ^1^Н- and ^13^С nuclear magnetic resonance (NМР). There was found to be 0.1% of taraxasterol in air-dried flowers, 1.0% in leaves, and 0.1% in stems. Leaves were found to contain 0.08% of taraxasteryl acetate, stems 0.05%, flowers 0.3%, receptacles 2.8%, leaves of involucre 0.13%, pappus of achenes 0.7%, and roots 0.1%. Lupeol and α-amyrin and their acetates were found in fat oil of achenes, which was isolated by extraction with hexane [36,37,38,39].

### 5.3. Steroids

Nolasco [40] detected in achenes Δ^5^ avenasterol, campesterol, stigmasterol, β-sitosterol, brassicasterol, and cholesterol. Another work [41] described the presence in achenes of *O. acanthium* phytosterols such as derivative of 4-desmethylsterol (campesterol, stigmasterol, β-sitosterol, Δ^7^ sitosterol, Δ^7^ avenasterol). Authors also investigated the dynamics of phytosterol accumulation during three periods (days after flowering, DAF). The first period included 5–14 DAF, the second period 14–38 DAF, and the third period 38–45 DAF. Stigmasterol in the amount of 10.48 ± 1.47% from all sterols was accumulated at 14 DAF, followed by campesterol (10.37 ± 0.36% at 14 DAF), stigmastanol (4.86 ± 0.22% at 38 DAF), and Δ^7^ sitosterol (10.43 ± 0.51% at 45 DAF). The main component was β-sitosterol (77.79 ± 0.4% in all sterols or 632 g/kg air-dried achenes). The total amount of sterols in achenes was 25 g/kg, and 5 g/kg in oil [42].

In fatty oil, free sterols were found: β-sitosterol, campesterol, traces of sitosterol and stigmasterol [39]. In roots, stigmasterol, β-sitosterol, and 24-methylenecholesterol were found (Table 4) [24,25,31].

### 5.4. Polyacetylenes

Bohlmann [43] detected polyacetylenes in the roots of *O. acanthium* (Table 5).

### 5.5. Fatty Acids

Ul’chenko [39,44,45,46,47,48] isolated fatty oil from 100 g of achenes by extraction with hexane at room temperature. Substances from the husks of cypselae were extracted by petroleum ether (40–60 °C). The division on fractions was performed in a column with silica gel. There were nine fractions of glycerids, with the main fraction consisting of triglycerides (88.0%). Alkaline hydrolysis with potassium hydroxide in methanol and enzyme hydrolysis with pancreatic lipase were carried out. To detect fatty acids and triglycerid transesterification in methanol, gas-liquid chromatography was carried out. The chemical structure of lipophilic components was identified with infrared (IR)- and NMR spectrometry.

Arfaoui [41] investigated the dynamics of fatty acid accumulation in achenes during three phases. The main fatty acids in oil from *O. acanthium* are linoleic, oleic, palmitic, stearic, and pentadecanoic acids. In contrast, epoxy-, hydroxy acids were not found in the work of Ul’chenko et al. The total amount of saturated and unsaturated acids in mature seeds was found to be 0.11–2.23 mg/g.

Matthaus [49] collected seeds of *O. acanthium* in Konya Province in Turkey. Mature seeds contained 15.71% of crude oil. The main fatty acids were linoleic (65.9%), oleic (18.8%), palmitic (5.8%), and stearic (2.6%) acids.

Zhelev [42] investigated fatty oil of *Carduus thoermeri* Weinm., *O. acanthium* L. and *Silybum marianum* L. from Bulgaria. The main fatty acids in the oil of *O. acanthium* were linoleic (576.5 g/kg), oleic (287.9 g/kg), palmitic (88.1 g/kg), and stearic (39.5 g/kg) acids. In comparison with this, the work of Tonguç [50] described a smaller amount of fatty acids (212 g/kg vs. 144 g/kg). Additionally, linoleic and linolenic acids were detected in the leaves of *O. acanthium* [35].

In those works, epoxy acids such as vernolic and coronaric acids were identified; saturated acids such as hentriacontanoic, nonacosanoic, stearic, palmitic, arachidic, pentadecanoic, margaric, myristic, and behenic acids; unsaturated acids with one double bond such as palmitoleic, oleic, gadoleic, erucic, and vaccenic acids; and polyunsaturated acids such as linoleic acid. Furthermore, 13-oxo-9Z,11E-octadecadienoic acid was found in roots [24,31].

### 5.6. Tocols

Matthaus [49] found tocopherols in fatty oil of achenes of *O. acanthium;* present was α-tocopherol (24.7 mg/100 g of oil); α-tocotrienol (1.1 mg/100 g of oil), β-tocopherol (0.4 mg/100 g of oil), and γ-tocopherol (0.2 mg/100 g of oil). Moreover, Zhelev [42] detected α-tocopherol (911 g/kg of oil), α-tocotrienol (89 g/kg of oil).

### 5.7. Nitrogen-Containing Compounds

Leaves of *O. acanthium* were found to contain 0.05% of alkaloids, and achenes were found to contain 0.1% of alkaloids [51]. In herb, 1-amino-2-propanol [52] and stachydrine [27] were found. Furthermore, in flowers, choline was detected [23]. The composition of amino acids was also determined [23].

### 5.8. Other Components

Muhamedov [53] determined the content of phospholipids and phytin in seeds of *O. acanthium*. Seeds were degreased with acetone, and phospholipids were extracted by chloroform–methanol (2:1). Dried extract was purified with acetone. The residue in the extract was dissolved in a mixture of chloroform–methanol–water (90:10:1), and then phospholipids were purified by size-exclusion chromatography and weighed. Quality analysis of phospholipids was carried out by thin layer chromatography (TLC). Phytin was extracted by 1% nitric acid from dried plant material. Thus, there was found to be 0.8% of phospholipids and 3.6% of phytin [53].

Qaderi [54] isolated an autoinhibitor of cypsela germination, a derivative of benzamide.

The content of phenols in air-dried achenes, which were extracted by methanol–water, was 2740 ± 26 mg of gallic acid/100 g of achenes [55].

## 6. Pharmacological Research

Extracts and individual substances from different parts of *O. acanthium* were investigated in vitro, in vivo, and in silico.

### 6.1. In Silico

Sharifi [20] isolated an inhibitor of ACE from achenes of *O. acanthium,* (E)-1-oxo-3,4-dihydro-1 H-isochromen-7-yl 3-(3,4-dihydroxyphenyl) acrylate as a yellow powder in the amount of 70 mg from 100 g of achenes. Activity of this compound was detected on the substrate hippuryl-L-histidyl-L-leucine (HHL). ACE hydrolyzes HHL on hippuric acid (HA). The amount of HA was measured by reversed-phase high-performance liquid chromatography (RP-HPLC). Inhibitor activity was measured based on the area under curve (AUC) of HA peak and expressed as ACE inhibition (%). The activity of the compound at the concentration of 330 μg/mL was 83 ± 1%. IC_50_ was 300 ± 25 μM as measured using a dose-response curve by nonlinear regression. Pharmacokinetics were determined by the absorption, distribution, metabolism, and excretion (ADME) method in Accelrys Discovery Studio 2.1 (BIOVIA, San Diego, CA, USA). The compound is highly soluble in water, has low blood–brain-barrier penetration, is well absorbed in the intestine, has more than 95% of association with plasma proteins and does not inhibit cytochrome P450. Molecular docking was performed in AutoDock4.2 (The Scripps Research Institute, La Jolia, CA, USA). There were C- and N- domains of ACE such as targets from the RCSB (the Research Collaboratory for Structural Bioinformatics) Protein Data Bank. The lowest binding energy of the C-domain was −8.31 kcal/mol, and the lowest binding energy for the N-domain was −8.29 kcal/mol.

The compound was named “*onopordia*”. Furthermore, by the method of fluorescence in vitro, the researchers tested onopordia specific activity against ACE domains; a pharmacophoric model of interaction between onopordia and the N-domain was created for the screening of new ACE inhibitors. Fluorescent assay was used for measuring ACE inhibitor activity. ACE hydrolyzes substrates Abz-SDK (Dnp)P-OH (o-Aminobenzoic acid-Ser-Asp-Lys(DNP)P-OH trifluoroacetate salt) and Abz-LFK (Dnp)-OH (O-Aminobenzoic acid-Leu-Phe-Lys(DNP)-OH trifluoroacetate salt, which specifically interact with the N- and C-domain, respectively. When ACEs are inhibited, there are increased amounts of Ac-SDKP (N-acetyl-seryl-aspartyl-lysylproline). Additionally, there are decreased fluorescent by-products of hydrolyze ACE. IC_50_ was measured for the C-domain (244 ± 9.0 μM) and for the N-domain (180 ± 1.8 μM). The results were compared with those for captopril (IC_50_ 2.81 ± 0.07 nM for the C-domain and 0.9 ± 0.06 nM for the N-domain). The presence of hydroxyl groups in the phenolic ring are the main markers for N-domain inhibitors. The acrylate double bond and the isochroman-1-one ring are the main markers for C-domain inhibitors [55,56].

### 6.2. In vitro

Extracts and individual compounds present anti-inflammatory, antiradical, antiproliferative, and antibacterial activities.

#### 6.2.1. Anti-Inflammatory Activity

The anti-inflammatory action of hexane and methanol extracts from achenes described below should be connected with the presence of arctiin and isochlorogenic acid [29]. Hexane and methanol extracts were identified. Paraffins were identified in hexane extract. Methanol extract was found to contain arctiin (arctigenin-4’-O-D-glucoside), arctigenin, and isochlorogenic acid by gas chromatography mass-spectrometry (GC-MS), high-performance liquid chromatography-mass spectrometry (HPLC-MS), high-performance liquid chromatography-ultraviolet (HPLC-UV) and proton nuclear magnetic resonance (^1^H-NMR) methods. Anti-inflammatory activity was detected on the immortalized human umbilical vein endothelial cells (HUVECtert). These cells were stimulated by lipopolysacharids and tumor necrosis factor-alpha (TNF-α). Decreased secretion of E-selectin by HUVECtert was measured by real-time quantitative reverse transcription polymerase chain reaction (RT-PCR) and decreased production of interleukin-8 (IL-8) was measured by enzyme-linked immunosorbent assay (ELISA). Furthermore, inhibitory activity of methanol extract on the secretion of E-selectin was detected, as was inhibitory activity of methanol and hexane extracts on the expression of E-selectin. No anti-inflammatory activity was detected by extracts when inflammation was stimulated by TNF-α. Methanol extract at a dose of 15–40 mg/mL or arctiin at a dose of 20–75 μM demonstrated inhibitory activity on E-selectin. The positive control was (E)-3-(4-Methylphenylsulfonyl)-2-propenenitrile (BAY 11-7082, an inhibitor of cytokine-induced IκB-α phosphorylation) at a dose of 5 μM.

Lajter [24] investigated the anti-inflammatory activity of lignans, flavonoids, and sesquiterpene lactones from the aerial part and roots of *O. acanthium* on different targets such as activity on the expression of cyclooxygenase-2 (COX-2) and nuclear factor kappa-light-chain-enhancer of activated B cells (NF-κB1), and its inhibitory effect on the production of NO, 5-lipoxygenase (5-LOX,) COX-1, and COX-2. The positive controls were quercetin (25 μM) and dexamethasone (2.5 nM) in the analysis of expression of NF-κB1 and COX-2; N^G^-monomethyl-L-arginine (L-NMMA) as inhibitor of NO-syntase (100 μM); zileuton as inhibitor of 5-LOX (5 μM); indomethacin and NS-398 (N-[2-(Cyclohexyloxy)-4-nitrophenyl]methanesulfonamide) as inhibitors of COX-1 and COX-2 (1.25 μM and 5 μM, respectively). Also, hexane, chloroform, and aqueous-methanol extracts from aerial parts and roots were investigated at a dose of 10 μg/mL (inhibition of COX-2 and NF-κB1 expression and NO-syntase) and 50 μg/mL (inhibition of 5-LOX, COX-1, and COX-2). Individual compounds were investigated at a dose of 20 μM dissolved in dimethyl suldoxide (DMSO). The results are presented as inhibition % ± SD (Table 6).

The compounds of roots 4β,15-Dihydro-3-dehydrozaluzanin C and zaluzanin C were also tested for inhibitory activity of the expression of COX-2 and NF-κB1. The most active on the expression of NF-κB1 was found to be 4β,15-Dihydro-3-dehydrozaluzanin C (inhibition of 96.0% for a dose of 10 μM), which was also found to be the most active inhibitor of COX-2 (inhibition of 91.2% on dose 10 μM). Zaluzanin C was demonstrated to have an inhibition of 83.7% for the expression of NF-κB1 and an inhibition of 87.9% for the expression of COX-2. The positive controls were found to have an inhibition of 47.6% for the expression of COX-2 (dexamethasone at a dose of 2.5 nM) and an inhibition of 46.0% for the expression of NF-κB1 (quercetin at a dose of 25 μM). In XTT viability assay (colorimetric assay by using 2,3-bis-(2-methoxy-4-nitro-5-sulfophenyl)-2H-tetrazolium-5-carboxanilide as a substrate which is reduced to purple formazan in living cells) at different concentrations at four, 24, 48 and 72 h), these compounds were demonstrated to have low cytotoxicity [24].

*Onopordopicrin* decreases intestinal inflammation [33]. Zaluzanin C and estafiatone showed an inhibitory effect on the lipopolysaccharide (LPS)/interferon-γ (INF-γ)-induced nitric oxide (NO) and prostaglandin E2 (PGE_2_) production on the RAW 264.7 macrophages with IC_50_ of 6.61 μM and 3.80 μM, respectively. Additionally, suppression of expression inducible NO syntaze (iNOS) and COX-2 was detected. Zaluzanin C inhibits electron and proton transport and inhibits the synthesis of ATP [33].

#### 6.2.2. Antitumor Activity

Abuharfeil [57] investigated aqueous extract of leaves and stems of *O. acanthium* against YAC cells (virus-induced murine T-cell lymphoma). The extract increased natural killer cell (NK) activity of splenic lymphoid cells, which exhibit a cytotoxic effect on YAC cells. There were 13 aqueous extracts. Extract from *O. acanthium* showed medium stimulation activity. The activity of aqueous extract from fresh material was 40.6% (ratio of effector:target 200:1), and that of aqueous extract from air-dried material was 30.4%. The positive control was interferon α at a dose of 500 U/mL, which increased the activity of NK cells by 78.5%.

Csupor-Löffler [58] investigated the antitumor activity of aqueous and organic extracts of 26 plants belonging to tribes Cynareae and Lactuceae (Asteraceae) on cell cultures: HeLa (cervix epithelial adenocarcinoma), A431 (skin epidermoid carcinoma), and MCF7 (breast epithelial adenocarcinoma). Living cells were detected in colorimetric MTT assay using [3-(4,5-dimethylthiazol-2-yl)-2,5-diphenyltetrazolium bromide. There were water, n-hexane, chloroform, and aqueous methanol extracts from mixture of flowers and fruits, leaves, and roots of *O. acanthium*. The most active was chloroform extract from leaves at a dose of 30 μg/mL (HeLa: IC_50_ 6.53 μg/mL, MCF7: IC_50_ 6.39 μg/mL, А431: IC_50_ 4.54 μg/mL) and roots (HeLa: IC_50_ 6.11 μg/mL, MCF7: IC_50_ 4.39 μg/mL, А431: IC_50_ 10.32 μg/mL). The positive controls were doxorubicin (half maximal effective concentration (EC_50_) HeLa: 0.081 μg/mL; MCF7: 0.152 μg/mL; А431: 0.081 μg/mL), cisplatin (IC_50_ HeLa: 3.7 μg/mL; MCF7: 2.8 μg/mL; А431: 0.85 μg/mL). The most active compounds from *O. acanthium* roots, 4β,15-Dihydro-3-dehydrozaluzanin C and zaluzanin C [24], were measured for their cytotoxic activity by XTT viability assay. The positive control was vinblastine (0.1 μg/mL). In the XTT viability assay THP-1 cells (human monocytic cell line) were used. These compounds had no or low effect on cell line.

Methanol extracts from leaves of *O. acanthium* and flowers of *Spartium junceum* L. (Fabaceae) were compared to each other on the human glioblastoma U-373 cell line. Methanol extract from *O. acanthium* was measured to have inhibitor activity of IC_50_ 309 μg/mL, while methanol extract from *S. junceum* was found to have inhibitor activity of IC_50_ 1602 μg/mL. This activity was measured by the trypan blue exclusion test. The mechanism of cytotoxicity is connected with inducing caspase-3, a cell enzyme of apoptosis. Thus, methanol extract of *O. acanthium* leaves showed proapoptotic activity [59].

Molnar [60] connected the antitumor activity of extracts of *O. acanthium* with effect on apoptotic enzymes. Sesquiterpene lactones from herb of *Artemisia asiatica* and roots of *O. acanthium* were investigated on the HL-60 cells (human leukemia cell). The compound from roots of *O. acanthium* 4β,15-dihydro-3-dehydrozaluzanin C demonstrated inhibitor activity of IC_50_ 3.6 μM. Cytotoxicity was measured by colorimetric MTT assay, and effects on the cell cycle were investigated by flow cytometric analysis. Cell changes were detected by staining cell DNA with the fluorescent marker propidium iodide and Hoechst 33258 (2′-(4-hydroxyphenyl)-5-(4-methyl-1-piperazinyl)-2,5′-bi-1H-benzimidazole trihydrochloride hydrate). The mechanism of cytotoxicity was inducing mitochondrial pathway of apoptosis by activation of caspase-3 and caspase-9.

The activity of caspase-3 was measured by fluorimetric assay using Ac-DEVD-AMC (N-Acetyl-Aspartyl-Glutamyl-Valyl-Aspartyl-7-amino-4-methylcoumarin) as a substrate. The activity of caspase-9 was measured using Ac-LEHD-pNA (N-Acetyl-Leucyl-Glutamyl-Histidyl-Aspartyl-p-Nitroanilide) as a substrate.

The lactones increased the amount of cells in G1 and G2/M populations, and decreased the amount of cells in the S population. After 48 h of incubation, an increased number of the hypodiploid (subG1) population was also detected for 5 and 10 μM doses. However, 4β,15-dihydro-3-dehydrozaluzanin C showed the lowest activity on this population. Compound 4β,15-dihydro-3-dehydrozaluzanin C demonstrated inducing activity on the caspase-3 and caspase-9.

Natural compounds that have α-methylen-γ-lactone in their structure may be potential antitumor agents.

NK activity was measured for human blood cells against K562 cells (human immortalized myelogenous leukemia cell line). The activity was tested after treatment with plant extract in the presence of IL-2 (1000 U/mL) and in the absence of IL-2. Production of cytokines IFNγ and TNFα was measured by ELISA; the activity of the cell enzymes granzyme A (N-alpha-benzyloxycarbonyl-L-lysine thiobenzyl ester (N-BLT) serine esterase) and N-acetyl-β-D-glucosaminidase (NAG) was measured by spectrophotometric analysis at 405 nm. In the presence of water extract from leaves and stems of *O. acanthium*, cytotoxicity at a dose of 0.1 mg/mL (dilution 1:50) of NK cells against K562 cells was 41.3% in the ratio effector:target of 200:1. In the presence of IL-2 this activity was 56.5%. Synthesis of cytokines after treatment with 0.1 mg/mL of water extracts in the presence of IL-2 was increased by 60% for TNFα and 87% for IFNγ. The activity of the cell enzymes granzyme A and NAG was measured only for water extracts from seeds of *Nigella sativum* [61].

Sesquiterpene lactones also demonstrate antitumor activity [33]. Thus, *onopordopicrin* has cytotoxic activity against the KB cell line (HeLa derivative). Onopordopicrin has an antiproliferative effect with an IC_50_ of 15 μM on the HL60 (promyelocytic leukemia). Cytotoxic effect was detected against cell lines P388 (murine leukemia), A549 (adenocarcinomic human alveolar basal epithelial cells), and HT29 (human colon cancer cell line). This effect connects with the presence of the α-methylene-γ-lactone group. Cytotoxic activity is increased by the presence of additional α,β-unsaturated ester group.

*Arctiopicrin* has cytotoxic activity on cell line of the colon cancer cell line in MTT colorimetric experiment. Toxic effects were investigated by brine shrimp lethality assay.

*Zaluzanin C* showed a cytotoxic effect on cell lines such as A549, SK-OV-3 (ovarian cancer cell line), SK-MEL-2 (human melanoma cell line), XF498 (human central nervous system (CNS) cancer), HCT15 (human colon adenocarcinoma), HL60, P388, HepG2 (human liver cancer cell line), HeLa, and OVCAR-3 (human ovarian carcinoma cell line). The effects of this compound on peptide synthesis inhibit the translocation of peptidyl-tRNA.

*Zaluzanin C* and *estafiatone* inhibit the proliferation of T- and B lymphocytes of mice with dose of 1 × 10^−5^ М or lower.

#### 6.2.3. Antiradical Activity

There are different data about the antioxidant activity of *O. acanthium*.

Kiselova [62] tested 23 aqueous plant extracts; for *O. acanthium,* air-dried flowers were used. In water extract, amounts of polyphenols were measured as µM quercetin equivalents. Antioxidant activity was measured by ABTS (2,2′-azinobis(3-ethylbenzothiazoline-6-sulfonic acid)) cation radical decolorization assay. In the presence of potassium persulfate, ABTS radical has absorbance at 734 nm that decreased if water extract contains some antioxidants. Flowers of *O. acanthium* demonstrated low antioxidant activity: 0.44 ± 0.06 mM Trolox equivalent antioxidant capacity (TEAC). The highest activity was 4.79 ± 0.14 mM TEAC by water extract from herb of *Alchemilla vulgaris* L.; the lowest activity was 0.31 ± 0.01 mM TEAC by water extract from flowers of *Calendula officinalis* L. The positive controls were the plants *Camellia sinensis* (L.) Kuntze, *Ilex paraguariensis* A.St.-Hil., *Aspalathus linearis* (Burm.f.) R.Dahlgren, and *Cyclopia intermedia* E.Mey.

Sharifi [20] carried out experiment using 2,2-diphenyl-1-picrylhydrazyl (DPPH). DPPH assay was based on the decreasing absorbance of DPPH solution at 517 nm then DPPH connection with radicals. Methanol extract of achenes of *O. acanthium* was demonstrated to have IC_50_ 2.6 ± 0.04 μg/mL. The positive controls were 6-hydroxy-2,5,7,8-tetramethylchroman-2-carboxylic acid (Trolox) and butylated hydroxytoluene (BHT), which had IC_50_ 3.3 ± 0.06 μ/mL and IC_50_ 10.3 ± 0.15 μ/mL, respectively.

In [55], the antioxidant activity of water-methanol extracts from air-dried achenes of *O. acanthium* was found to be IC_50_ 7.0 ± 0.09 μg/mL. The positive controls were BHT (IC_50_ 10.3 ± 0.15 μ/mL) and Trolox (IC_50_ 3.2 ± 0.06 μ/mL).

Koc [21] measured the antioxidant activity of methanol, ethanol, and acetone extracts from the flowers and leaves of *O. acanthium* using DPPH. The total phenolic content was investigated using Folin–Ciocalteu’s reagent, which consists of hexavalent phosphomolybdic/phosphotungstic acid complexes. This reagent is connected with hydroxyl groups of phenols and products have absorbance at 765 nm. Furthermore, authors researched the inhibitor effect on the antioxidant enzymes such as catalase (CAT), glutathione S-transferase (GST), and glutathione peroxidase (GPx). Free-radical scavenging activity was measured as IC_50_ and was calculated from the dose–response inhibition curve. Acetone extract had IC_50_ 842 ng/mL, ethanol extract IC_50_ 1120 ng/mL, and methanol extract IC_50_ 723 ng/mL. The positive controls were ascorbic acid, with IC_50_ 5.144 μg/mL, and quercetin, with IC_50_ 1.685 μg/mL. Extracts from *O. acanthium* materials did not show high inhibitory activity of antioxidant enzymes.

Similar research was conducted by Habibatni [22], using DPPH and butanol extract from leaves of *O. acanthium*. Antioxidant activity was measured as IC_50_ 134.4 mg/mL. The positive control was ascorbic acid (IC_50_ 21.4 μg/mL) and the negative control was a solution of DDPH in 80% methanol without *O. acanthium* extract. Additionally, authors investigated the hypouricemic effect as an inhibitor of xanthine oxidase. This experiment was carried out using the absorbance of products of xanthine and xanthine oxidase (the dose of xanthine oxidase was 0.1 U/mL). The positive control was allopurinol. The activity of butanol extract was 7.0% of inhibition at a dose of 100 μg/mL, and its IC_50_ was 572.9 μg/mL, while allopurinol had IC_50_ 3.9 μg/mL.

#### 6.2.4. Antibacterial Activity

Methanol extract from the aerial part of *O. acanthium* at a concentration of 100 µg/mL had no antibacterial effect on several cultures: *Klebsiella pneumonia*, *Proteus vulgaris*, *Shigella sonnei*, *Vibrio cholerae*, *Escherihia coli*, *Staphylococcus aureus*, *Bacillus anthracis*, and *Salmonella paratyphi A* [63].

The antibacterial effect of methanol extract from achenes of *O. acanthium* was investigated using the gram-negative cultures *Escherichia coli* and *Klebsiella pneumonia* and the gram-positive cultures *Staphylococcus epidermidis, S. aureus,* and *Micrococcus luteus* by agar disc diffusion method. The reference control was amikacin at a dose of 30 mg. The highest antibacterial effects were on the culture of *M. luteus,* with a diameter of zone inhibitor of 21 ± 1 mm, and on the culture of *S. epidermidis* (18.66 ± 1.53 mm). There was no inhibitory activity against *S. aureus*, *K. pneumonia*, or *E. coli*. Minimum inhibitory concentration (MIC) was measured by the broth macro dilution method and resazurin microtiter method. For the second method, kanamycin was used as the positive control. MICs were obtained for *S. epidermidis* and *M. luteus* (0.612 mg/mL). The total phenolic content was found to be 168.56 ± 4.89 mg of gallic acid/g of extract [64].

Ethyl acetate extract from fresh leaves of *O. acanthium* was investigated for antibacterial activity after dissolution in 10-times crude extract in ethanol. The extract was separated by TLC and offline overpressure layer chromatography (OPLC) direct bioautography. The most active fractions were analyzed by HPLC-MS. The structure of isolated compounds was detected by NMR. The extract was separated by low-pressure flash chromatography. This work used *Allivibrio fischeri*, *Bacillus subtilis* strain F1276, *Pseudomonas syringae* pv. *maculicola*, *Xanthomonas euvesicatoria*, *Lactobacillus plantarum*, *S. aureus*, methicillin resistant *S. aureus* (MRSA), and *E. coli.* The MIC was also measured for extracts between 27.6–275.9 μg/mL and for isolated onopordopicrin between 2.2–172.4 μg/mL by a broth microdilution method. In the results after TLC analysis, inhibitor activity of growth *B. subtilis* was found. There were two zones, one with a retardation factor (hR_f_) of 37 and one with an hR_f_ of 82. The zone with hR_f_ 82 was active against *B. subtilis*, *X. euvesicatoria*, *A. fischeri*, *S. aureus*, and MRSA. All of the bacterial cultures were sensitive to component hR_f_ 37; however *P. maculicola* was inhibited by a high amount of this compound. Onopordopicrin inhibited all of the bacterial cultures; however *P. maculicola* was resistant to 20 μg onopordopicrin [35].

*Estafiatone* has antibacterial activity against *B. subtilis*. *Zaluzanin C* inhibits root and shoot growth in lettuce, tomatoes, and cress, and has antifungal activity. Additionally, *onopordopicrin* demonstrated antibacterial activity against *S. aureus* [33].

#### 6.2.5. Antihypertensive Activity

In Iran, the screening of 50 plants was carried out in the presence ACE-inhibitor activity [55]. Air-dried achenes of *O. acanthium* (1 g) were extracted by 20 mL of methanol–water (80:20 v/v) at room temperature for 24 h followed by 2 h in an ultrasonic bath. The extract was filtered, concentrated under low pressure at room temperature, and then lyophilized. ACE-inhibitor activity was detected using HHL as a substrate. After hydrolyzing HHL by ACE, hippuric acid (HA) is produced. The amount of HA was measured by RP-HLPC. Inhibitor activity for the extract from *O. acanthium* achenes was 80.2 ± 2.0%.

### 6.3. In vivo

#### 6.3.1. Anti-Inflammatory Activity

Ivanova [65] investigated the activity of ethanol extract from herb of *O. acanthium* using two models: cotton pellet-induced granuloma model and caoline-induced paw edema model. In the first model, cotton pellets were implanted in the backs of rats, and removed after eight days. The increase in the dry weight of the pellets was taken as a measure of granuloma formation. In the caoline-induced model, animals were given subplantar injection of 0.1 mL 10% caoline. The raw volume was measured four hours, one day, and two days after the injection. Ethanol extracts at doses of 0.25 and 0.84 mg/kg were input orally, while the control group received saline; the negative control was 40% ethanol at a dose of 0.25 mL/kg and 0.84 mL/kg. In the cotton pellet-induced granuloma model, extract at the dose of 0.25 mg/mL decreased edema in the exudative and proliferative steps, while at the dose of 0.84 mg/mL the main effect was in the exudative stage. In the caoline model, the effect of the dose of 0.84 mg/mL was higher than that for the dose of 0.25 mg/mL.

Anti-inflammatory activity may be related to the presence of triterpenoids. Taraxasterol and taraxasteryl acetate were extracted by chloroform from leaves and flowers of *O. acanthium* and were then separated by column chromatography using silica gel. Taraxasterol and its derivatives showed activity in formaline-induced edema model albino rats. Percentile edema inhibition was measured in terms of paw thickness. Inflammation was induced by subplantar injection of 0.1 mL of 3% formaline. The control group was orally administered saline 30 min before the injection. The positive control was prednisolone administered orally at a dose of 10 mg/kg 30 min before formaline injection. Taraxasterol was administered orally at a dose of 10 mg/kg 30 min before the formaline injection. In the results, taraxasterol was found to decrease inflammation by 38.2% after three hours and by 31.4% after 24 h. Meanwhile, prednisolone decreased inflammation by 35.3% after three hours and by 32.6% after 24 h. In the control group, inflammation decreased by 41.2% after three hours and by 41.8% after 24 h [66].

Habibatni [22] investigated extracts of methanol, hexane, chloroform, ethyl acetate, and butanol from *O. acanthium* leaves for cytotoxicity using *Artemia salina* leach. The lowest cytotoxic effect was observed for the butanol extract. The anti-inflammatory activity of the butanol extract was researched by carrageenan-induced edema model on four groups of six Wistar rats. The positive control received 100 mg/mL of aspirin orally one hour before an injection of 0.1 mL of 0.6% carrageenan. The negative control received 0.9% saline orally one hour before an injection of 0.1 mL of 0.6% carrageenan. In the results, inflammation was inhibited by 37.78% by a dose of 200 mg/kg of butanol extract and by 40.52% by a dose of 400 mg/kg of butanol extract. Aspirin at a dose of 100 mg/kg decreased inflammation by 42.62%.

#### 6.3.2. Antipyretic Activity

The antipyretic effect of butanol extract of *O. acanthium* leaves has been tested on rats [22]. Animals received subcutaneous injection of a pyrogenic dose of brewer’s yeast 20% (1 mL/100 g body weight). The basal temperature was measured rectally every hour for five hours. The positive control was aspirin administered orally at a dose of 500 mg/mL and the negative control was water administered orally. Butanol extract was administered at doses of 200 and 400 mg/mL orally. The basal temperature was 36.38°C. After five hours of injection, the body temperature was 39.5°C in the control group. Aspirin decreased body temperature to 37.22°C. In the group treated with butanol extract at a dose of 200 mg/mL, body temperature was 37.38°C, while in the group treated with 400 mg/mL of butanol extract, body temperature was decreased to 37.34°C.

#### 6.3.3. Analgesic Activity

Analgesic activity was detected by abdominal cramps in mice 30 min after intraperitoneal injection of 1% acetic acid. The positive control was 500 mg/kg aspirin administered orally 30 min before the injection of acetic acid. The negative control was saline administered 30 min before injection of acetic acid. Butanol extract was administered at doses of 200 and 400 mg/kg 30 min before injection of acetic acid. The analgesic effect was measured as a percentage decrease in cramps. For the dose of 200 mg/kg of butanol extract, this decrease was 72.0%, while for the dose of 400 mg/kg it was 76.57%; in the positive group, the decrease in cramps was 86.29% [22].

#### 6.3.4. Toxicity

Female BALB/C mice were treated with water herbal extracts from 20 different plants instead of water for three weeks. Their blood was haematologically tested each week (haemoglobin, hematocrit, amount of white blood cells) and biochemical tests were conducted (glucose, urea, cholesterol, creatinine, direct and total bilirubin). All extracts were safe. Then, the activity of NK cells was measured during four weeks of oral treatment with the water extracts. Each week, the spleen was isolated and splenic lymphoid cells were tested for NK activity against YAC cells. In the results, water extract from fresh leaves and stems of *O. acanthium* induced NK activity after one week of treatment at 38.6% at an effector:target ratio of 200:1. For the mixture of water extracts with high and medium activity (seeds of *N. sativum,* leaves and stems of *O. Acanthium,* bulb of *Allium sativum* and *Allium cepa*), after one week of treatment the NK activity was increased by 65.3% at an effector:target ratio of 200:1 [67].

Additionally, toxicity was investigated in the male mice. Dry extract from achenes of *O. acanthium* was dissolved in DMSO and injected intraperitoneally. Animals’ condition (lethargy and movement) and animal death were investigated for 24–48 h. At a dose of 0.5–5 g/kg of animal weight, there was no change in the condition or deaths of mice. At a dose of 20 g/kg, all animals died. Lethal Dose, 50% (LD_50_) was 8.44 ± 0.04 g/kg. Extract from the cypsela of *O. acanthium* was described as “practically nonpoisonous” [68].

#### 6.3.5. Regeneration Effect

Hydroalcoholic extract from flowers of *O. acanthiumin* was used in in vivo experiments on female and male rabbits for the treatment of wounds. There were five groups of seven animals. Animals were shaved in the dorsal area. Then, a wound 2 × 2 cm in size was cut by a metal plate. Animals were anaesthetized by subcutaneous injection of 2 mL of 2% lidocaine. The area of the wound was measured by placing a transparent film over graph paper. The extract was dissolved to produce the 0.1–1% ointment with eucerin (lanolin, water, oil) as an ointment base. The ethanol extract was prepared from leaves, stems, flowers, and roots using 70% ethanol. The most effective was found to be flower extract administered at doses of 0.0025, 0.05, 0.1, 0.2, and 0.4% (w/w). The optimum concentration was 0.2% and the maximum regeneration was at six days after extract administration. The minimum time for 50% of wound treatment was 3.2 days. The control group was treated with eucerin and 50% of wound area was treated after 7.2 days [69].

#### 6.3.6. Antihypertensive Effect

Patients between 30–60 years in age were involved in an investigated group treated by capsules with 1 g of dry extract from seeds of *O. acanthium* from 11 g air-dried achenes extracted by ethanol (1:8), and the extract was evaporated and prepared in capsules. Patients with stage I–II hypertension (blood pressure higher then 140/90 mmHg) were treated with 50 mg/day losartan six weeks before the experiments. They were then administered with two capsules two times a day for eight weeks. Arterial blood pressure was measured each week and metabolic parameters (lipid profile, liver function tests, blood urea nitrogen, chromium, fasting blood sugar) were measured two times before and after treatment. In clinical experiments, the base systolic blood pressure of 151.9 ± 13.74 mmHg was decreased to 134.6 ± 18.25 mmHg at the end of the therapy. Diastolic blood pressure, which was originally 97.41 ± 10.36 mmHg, had decreased to 85.71 ± 7.481 mmHg at the end of therapy. There were no changes in the metabolic parameters [68].

A summary of pharmacological properties of extracts and individual compounds can be found in Table 7.

## 7. Conclusions

The traditional usage of the aerial part of *Onopordum acanthium,* which is an edible plant, a source of honey, and an anti-inflammatory, antitumor, and cardiotonic remedy, demonstrates the safety and non-toxicity of this plant. Research demonstrates that *O. acanthium* is a potential preventive and medicinal plant. Extracts from the herb, leaves, seeds, and individual substances such as sesquiterpene lactones, triterpenoids, lignans, etc., have anti-inflammatory, antiproliferative, and antihypertensive properties. These effects may be used in the treatment of chronic inflammation and to prevent various types of tumors. The presence of sesquiterpene lactones in the roots and the presence of triterpenoids in the aerial part correlate with antitumor and anti-inflammatory properties of extracts. More detailed study of chemical content, systematic in vitro and in vivo tests, and comparative study of *O. acanthium* and related species such as *O. illyricum* L., *Carduus nutans* L., and *Cirsium vulgare* (Savi) Ten. may be useful work.

## Figures and Tables

**Figure 1 plants-08-00040-f001:**
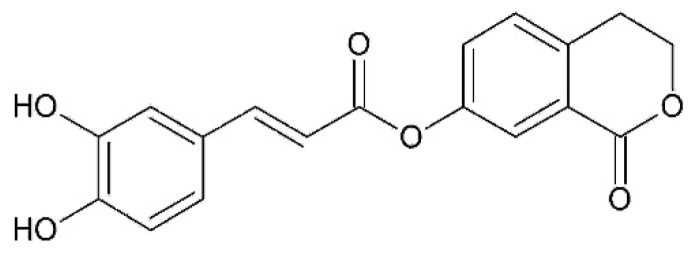
Inhibitor of angiotensin-converting enzyme (ACE) isolated from achenes of *Onopordum acanthium*, 1, (E)-1-oxo-3,4-dihydro-1 H-isochromen-7-yl 3-(3,4-dihydroxyphenyl) acrylate.

**Figure 2 plants-08-00040-f002:**
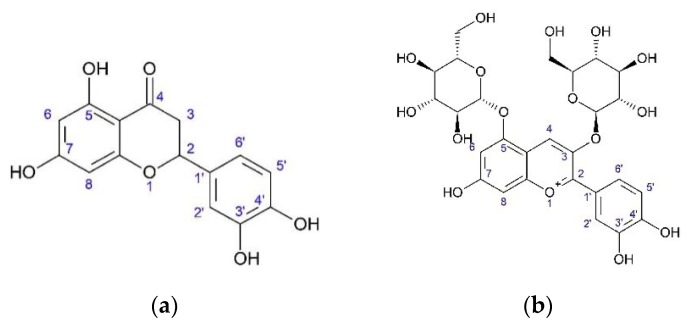
Flavanone and antocyanin from *O. acanthium* L.: (**a**) Eriodictyol; (**b**) Cyanin.

**Figure 3 plants-08-00040-f003:**
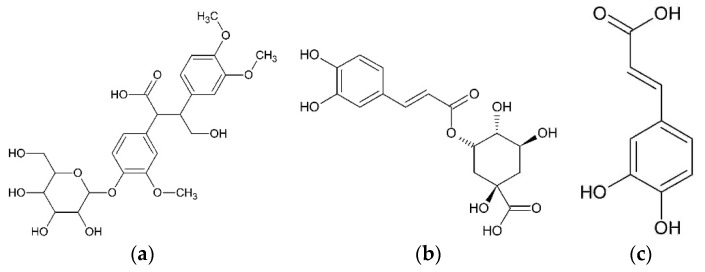
Phenylpropanoids from *O. acanthium* L.: (**a**) Aconiside; (**b**) Isochlorogenic acid; (**c**) Caffeic acid.

**Figure 4 plants-08-00040-f004:**
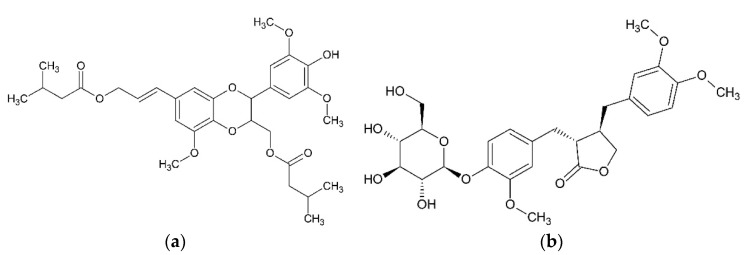
Lignans from *O. acanthium* L.: (**a**) Nitidanin diisovalerianate; (**b**) Arctiin.

**Figure 5 plants-08-00040-f005:**
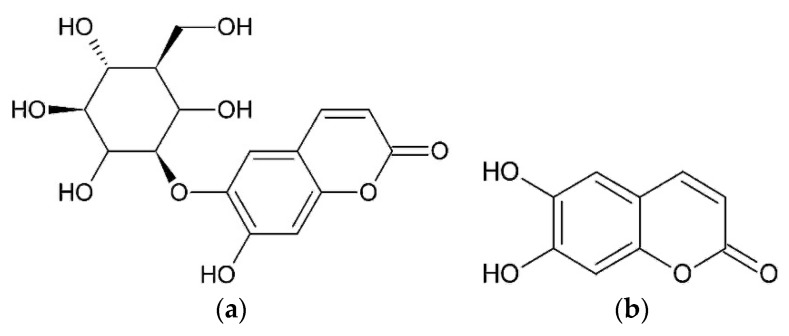
Coumarins from *O. acanthium* L.: (**a**) Aesculin; (**b**) Aesculetin.

**Figure 6 plants-08-00040-f006:**
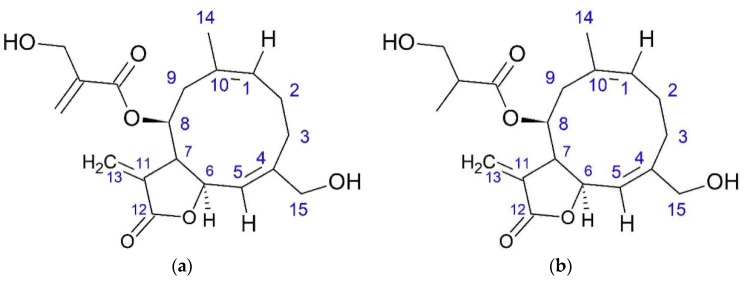
Germacranolides from *O. acanthium* L.: (**a**) Onopordopicrin; (**b**) Arctiopicrin.

**Figure 7 plants-08-00040-f007:**
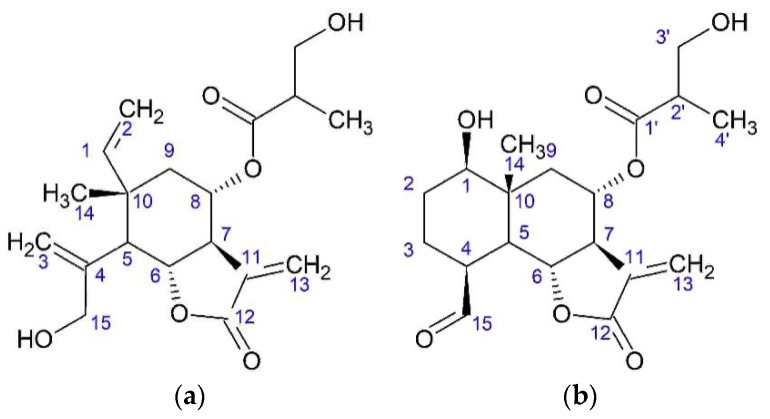
Elemanolide and eudesmanolide from *O. acanthium* L.: (**a**) Elemanolide 11(13)-dehydromelitensin β-hydroxyisobutyrate; (**b**) Acanthiolide.

**Figure 8 plants-08-00040-f008:**
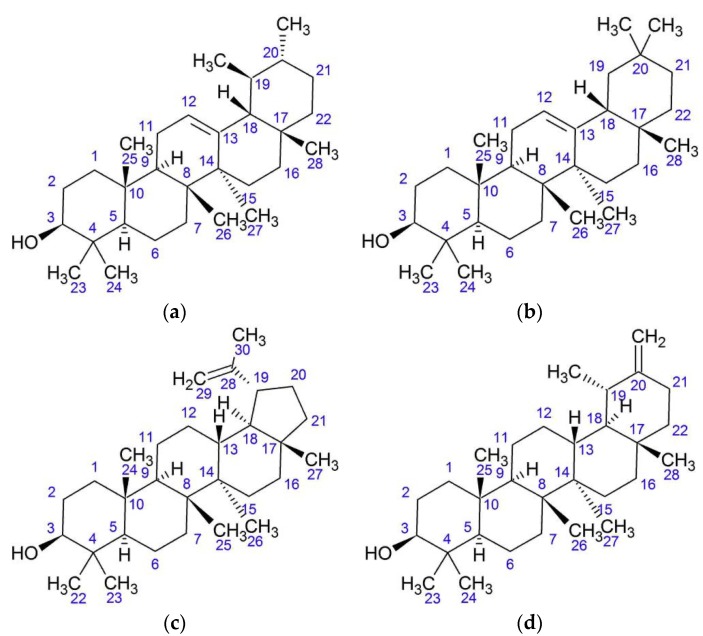
Triterpenoids from *O. acanthium* L.: (**a**) α-amyrin; (**b**) β-amyrin; (**c**) Lupeol; (**d**) Taraxasterol.

**Table 1 plants-08-00040-t001:**
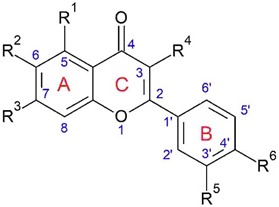
Flavonoids from *Onopordum acanthium* L.

No	Name	R^1^	R^2^	R^3^	R^4^	R^5^	R^6^
1.	Apigenin	OH	H	OH	H	H	OH
2.	Luteolin	OH	H	OH	H	OH	OH
3.	Scutellarein	OH	OH	OH	H	H	OH
4.	Nepetin	OH	OCH_3_	OH	H	OH	OH
5.	Chrysoeriol	OH	H	OH	H	OCH_3_	OH
6.	Hispidulin	OH	OCH_3_	OH	H	H	OH
7.	Pectolinarigenin	OH	OCH_3_	OH	H	H	OCH_3_
8.	Scutellarein 4’-methyl ether	OH	OH	OH	H	H	OCH_3_
9.	Quercetin	OH	H	OH	OH	OH	OH
10.	Apigenin-7-O-glucoside	OH	H	O-glucose	H	H	OH
11.	Apigenin-7-O-rutinoside	OH	H	O-rutinose	H	H	OH
12.	Apigenin-7-O-β-D-glucuronide	OH	H	O-glucuronic acid	H	H	OH
13.	Luteolin-7-O-glucoside	OH	H	O-glucose	H	OH	OH
14.	Quercetin-3-O-glucoside	OH	H	OH	O-glucose	OH	OH
15.	Isorhamnetin-3-O-glucoside	OH	H	OH	O-glucose	OCH_3_	OH

**Table 2 plants-08-00040-t002:**
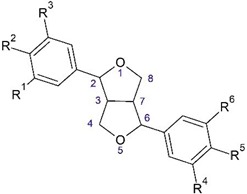
Bis-epoxy forms of lignans from *O. acanthium* L.

No	Name	R^1^	R^2^	R^3^	R^4^	R^5^	R^6^
1.	Pinoresinol	H	OH	OCH_3_	H	OH	OCH_3_
2.	Syringaresinol	OCH_3_	OH	OCH_3_	OCH_3_	OH	OCH_3_
3.	Medioresinol	OCH_3_	OH	OCH_3_	H	OH	OCH_3_

**Table 3 plants-08-00040-t003:**
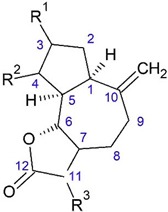
Guianolides from *O. acanthium* L.

No	Name	R^1^	R^2^	R^3^
1.	4β,15-dihydro-3-dehydrozaluzanin C	-HС(O)	α-CH_3_	CH_2_
2.	Zaluzanin C	OH	CH_2_	CH_2_
3.	4β,15,11β,13-tetrahydrozaluzanin C	β-OH	α-CH_3_	β-CH_3_

**Table 4 plants-08-00040-t004:**
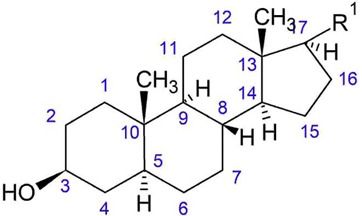
Steroids from *O. acanthium* L.

No	Name	R^1^	Δ
1.	Δ^5^ avenasterol	-CH(α-CH_3_)-CH_2—_CH_2_-C(CH(CH_3_)_2_)=CH-CH_3_	5
2.	Δ^7^ avenasterol	-CH(α-CH_3_)-CH_2—_CH_2_-C(CH(CH_3_)_2_)=CH-CH_3_	7
3.	Campesterol	-CH(α-CH_3_)-CH_2_-CH_2_-CH(α-CH_3_)-CH(CH_3_)-CH_3_	5
4.	Δ^7^ campesterol	-CH(α-CH_3_)-CH_2_-CH_2_-CH(β-CH_3_)-CH(CH_3_)-CH_3_	7
5.	Stigmasterol	-CH(α-CH_3_)-CH=CH-(α)CH(C_2_H_5_)-CH(CH3)-CH_3_	5
6.	Δ^7^ stigmasterol	-CH(α-CH_3_)-CH=CH-(α)CH(C_2_H_5_)-CH(CH_3_)-CH_3_	5, 7
7.	Stigmastanol	-CH(α-CH_3_)-CH_2_-CH_2_-(α)CH(C_2_H_5_)-CH(CH_3_)-CH_3_	-
8.	β-sitosterol	-CH(α-CH_3_)-CH_2_-CH_2_-(α)CH(C_2_H_5_)-CH(CH_3_)-CH_3_	5
9.	Brassicasterol	-CH(α-CH_3_)-CH=CH-CH(β-CH_3_)-CH(CH_3_)-CH_3_	5
10.	Δ^7^ sitosterol	-CH(α-CH_3_)-CH_2_-CH_2_-(α)CH(C_2_H_5_)-CH(CH_3_)-CH_3_	7
11.	Cholesterol	-CH(α-CH_3_)-CH_2_-CH_2_-CH_2_-CH(CH_3_)-CH_3_	5
12.	24-methylenecholesterol	-CH(α-CH_3_)-CH_2_-CH_2_-C(CH(CH_3_)_2_)=CH_2_	5

**Table 5 plants-08-00040-t005:** Polyacetylenes from *O. acanthium* L.

No	Name	Structure
1.	Heptadecatetraen-(2, 8, 10, 16)-diin-(4, 6)-al-(1)	(2E,8E,10E) CH(O)-CH=CH-(C≡C)_2_-(CH=CH)_2_-(CH_2_)_4_-CH=CH_2_
2.	Tridecadien-(1, 11)-tetrain-(3, 5, 7, 9)	CH_3_-CH=CH-(C≡C)_4_-CH=CH_2_
3.	Heptadecatetraen-(1, 7, 9, 15)-diin-(11, 13)	(7E,9E,15E) CH_3-_CH=CH-(C≡C)_2_-(CH=CH)_2_-(CH_2_)_4_-CH=CH_2_
4.	Heptadecatetraen-(2, 8, 10, 16)-diin-(4, 6)-ol-(1)	(2E,8E,10E) HO-CH_2_-CH=CH-(C≡C)_2_-(CH=CH)_2_-(CH_2_)_4_-CH=CH_2_

**Table 6 plants-08-00040-t006:** The anti-inflammatory activity of the extracts and compounds from aerial parts and roots of *O. acanthium* L. (Lajter [24]).

Parts of *O. acanthium*	Extracts/Compound	COX-2 ^2^	5-LOX ^3^	NO-syntase	COX-1 ^1^
		Inhibition (%)			
Aerial parts	Hexane	82.8			
	Aqueous-methanol		31.2		
	Chloroform			76.7	
	Pinoresinol			49.13	
	Medioresinol				16.2
	Syringaresinol		28.5		
	Hispidulin		51.6		
	Nepetin		62.4		
	Apigenin		41.3		
	Luteolin		74.6		
Roots	Hexane	86.5			
	Aqueous-methanol		59.7		
	Chloroform		56.7		
	4β,15-Dihydro-3-dehydrozaluzanin C			100.4	
	Zaluzanin C			99.4	
	4β,15,11β,13-Tetrahydrozaluzanin C			61.4	
	Nitidanin diisovalerianate		16.1		
	13-Oxo-9Z,11E-octadecadienoic acid		20.4		
	24-Methylenecholesterol	36.4			

^1^ Cyclooxygenase-1,^2^ Cyclooxygenase-2, ^3^ 5-lipoxygenase

**Table 7 plants-08-00040-t007:** Summary of the effects of extracts and individual compounds from *Onopordum acanthium* L.

Part of plant	Extract/compound	Activity	Method	Reference
Aerial part	hexane, aqueous-methanol, chloroform/pinoresinol, medioresinol, syringaresinol, hispidulin, nepetin, apigenin, luteolin	anti-inflammatory	activity on the expression of COX-2 and NF-κB1 ^1^ (real-time PCR ^2^), inhibitory effect on the production of NO (Griess assay method), 5-LOX (EIA ^3^), COX-1, and COX-2 (EIA ^3^)	[24]
	ethanol		cotton pellet-induced granuloma model and caoline-induced paw edema model per rats	[65]
Leaves	aqueous	antitumor, cytotoxicity	NK ^4^ activity of splenic lymphoid cells isolated from spleens of eight- to 10-week-old BALB/C female mice. Splenic cells provide cytotoxic effect on YAC ^5^ cells detected by ^51^Cr-release assay	[57]
			activity of NK ^4^ cells from human peripheral blood against K562 ^6^ cells by staining viable cells with neutral red solution; production of cytokines IFNγ ^7^ and TNFα ^8^ by ELISA ^9^	[61]
			the toxicity of aqueous extract was measured on eight- to 10-week-old BALB/C female mice. The blood was haematologically tested each week (haemoglobin, hematocrit, amount of white blood cells) and biochemical tests were performed (glucose, urea, cholesterol, creatinine, direct and total bilirubin). NK ^4^ activity of splenic lymphoid cells isolated from BALB/C mice spleen those were treated by plant extract for three weeks. Splenic cells have a cytotoxic effect on YAC ^5^ cells detected by ^51^Cr-release assay	[67]
	methanol		trypan blue exclusion test using cell culture U-373 ^10^	[59]
	chloroform		MTT ^11^ assay using cell cultures: HeLa ^12^, A431 ^13^, and MCF7 ^14^	[58]
		anti-inflammatory	formaline-induced edema model per albino rats	[66]
	methanol. ethanol, acetone	antiradical	DPPH ^15^ antioxidant assay	[21]
	ethyl acetate	antibacterial	gram-negative cultures *Allivibrio fischeri*, *Pseudomonas syringae* pv. *maculicola, Xanthomonas euvesicatoria, and Escherichia coli;* gram-positive cultures *Bacillus subtilis* strain F1276, *Lactobacillus plantarum*, *Staphylococcus aureus*, and methicillin resistant *S. aureus* (MRSA) by TLC ^16^ and OPLC ^17^ direct bioautography	[35]
	butanol	antiradical	DPPH ^15^ antioxidant assay	[22]
		anti-inflammatory	carrageenan-induced edema model on four groups per six Wistar rats	
		antipyretic	decrease of pyretic effect induced by injection of brewer’s yeast 20% to rats	
		analgesic	abdominal cramps in mice 30 min after intraperitoneal injection of 1% acetic acid	
Stems	aqueous	antitumor	NK ^4^ activity of splenic lymphoid cells isolated from spleens of eight- to 10-week-old BALB/C female mice. Splenic cells exert a cytotoxic effect on the YAC ^5^ cells detected by ^51^Cr-release assay	[57]
			activity of NK ^4^ cells from human peripheral blood against K562 ^6^ cells by staining viable cells with neutral red solution; production of cytokines IFNγ ^7^ and TNFα ^8^ by ELISA ^9^	[61]
			the toxicity of aqueous extract was measured on eight- to 10-week-old BALB/C female mice. The blood was haematologically tested each week at (haemoglobin, hematocrit, amount of white blood cells) and biochemical tests were conducted (glucose, urea, cholesterol, creatinine, direct and total bilirubin). NK ^4^ activity of splenic lymphoid cells isolated from BALB/C mice spleen those were treated by plant extract for three weeks. Splenic cells exert a cytotoxic effect on YAC ^5^ cells detected by ^51^Cr-release assay	[67]
Inflorescences	aqueous	antiradical	ABTS ^18^ antioxidant assay	[62]
	methanol. ethanol, acetone		DPPH ^15^ antioxidant assay	[21]
	ethanol	regeneration	0.2% ointment with base eucerin using female and male rabbits treated for wounds	[69]
Achenes	methanol/arctiin, isochlorogenic acid	anti-inflammatory	real-time PCR ^2^ and ELISA ^9^ using cell culture HUVECtert ^19^	[29]
	water-methanol/(E)-1-oxo-3,4-dihydro-1 H-isochromen-7-yl 3-(3,4-dihydroxyphenyl) acrylate (“onopordia”)	antiradical	DPPH ^15^ antioxidant assay	[20,55]
		inhibitor of angiotensin-converting enzyme (ACE)	RP-HLPC ^20^, molecular docking, fluorescent assay; treatment of 20 patients between 30–60 years in age with stage I–II hypertension by capsules with 1 g of dry extract for eight weeks	[20,55,68]
	methanol	antibacterial	gram-negative cultures *Escherichia coli* and *Klebsiella pneumonia* and gram-positive cultures *Staphylococcus epidermidis, Staphylococcus aureus,* and *Micrococcus luteus* byagar disc diffusion method	[64]
Roots	hexane, aqueous-methanol, chloroform/4β,15-Dihydro-3-dehydrozaluzanin C, zaluzanin C, 4β,15,11β,13-Tetrahydrozaluzanin C, nitidanin diisovalerianate, 13-Oxo-9Z,11E-octadecadienoic acid, 24-methylenecholesterol	cytotoxic, anti-inflammatory	XTT ^20^ viability assay, activity on the expression of COX-2 and NF-κB1 ^1^ (real-time PCR ^2^), inhibitory effect on the production of NO (Griess assay method), 5-LOX (EIA ^3^), COX-1 and COX-2 (EIA ^3^)	[24]
	chloroform	antitumor	MTT ^11^ assay using cell cultures: HeLa ^12^, A431 ^13^, MCF7 ^14^	[58]
	4β,15-dihydro-3-dehydrozaluzanin C		MTT ^11^ assay using cell culture HL-60 ^22^	[60]

^1^ Nuclear factor kappa-light-chain-enhancer of activated B cells,^2^ Polymerase chain reaction, ^3^ Enzyme immuno assay, ^4^ Natural killer cells, ^5^ Virus-induced murine T cell lymphoma, ^6^ Human immortalised myelogenous leukemia cell line, ^7^ Interferon gamma, ^8^ Tumor necrosis factor-alpha, ^9^ Enzyme-linked immunosorbent assay, ^10^ Human glioblastoma cell line, ^11^ [3-(4,5-dimethylthiazol-2-yl)-2,5-diphenyltetrazolium bromide, ^12^ Cervix epithelial adenocarcinoma, ^13^ Skin epidermoid carcinoma, ^14^ Breast epithelial adenocarcinoma, ^15^ 2,2-diphenyl-1-picrylhydrazyl, ^16^ Thin layer chromatography, ^17^ Offline overpressure layer chromatography, ^18^ 2,2′-azinobis(3-ethylbenzothiazoline-6-sulfonic acid, ^19^ Immortalized human umbilical vein endothelial cells, ^20^ Reversed-phase high-performance liquid chromatography, ^21^ 2,3-bis-(2-methoxy-4-nitro-5-sulfophenyl)-2H-tetrazolium-5-carboxanilide, ^22^ Human leukemia cells

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
