# Peer review of "Traditional Medicine Plant, *Onopordum acanthium* L. (Asteraceae): Chemical Composition and Pharmacological Research"

_plants, 2019, doi:10.3390/plants8020040_

Round 1
Reviewer 1 Report
The paper is a revision of chemical composition and pharmacological application of a traditional medicine plant, Onopordum AcanthiumL. (Asteraceae), the review is interesting and provides knowledge about the composition and health properties of a such known plant. However, there are some critical points that need to be clarified or corrected before publication:
- Table 1, I recommend to authors transform Table1 into Figure 2, to make the flavonoid structure bigger and the substituents table smaller. Additionally, I suggest to authors divided and put it in the horizontalorientation. Eriodictyol molecular structure would be Figure 3, and Cyanin would be Figure 4.
- Line 92, I think redaction of the sentence couldbe better like follow: The flavones, flavonols,andflavanonesaglyconshave been described in O. acanthium.
- Line 103, I suggest a new redaction for the sentence: Eriodictyol and quercetin were identified in flowers.
- Table 3, I recommend transforminginto figure follow the same structure I suggest to figure 2.
- Line 125, I think it would be interesting toexplain something about Sesquiterpenelactones.
- Figure 3, please to correct in the figure caption, the compounds are not lignans, they are coumarins.
- Table 5,6, 7, I recommend to reduce the size andto make more aesthetic.
- It is necessary a section of abbreviatures inorder to explainall the acronyms.
- Line 267, I think it is necessary a reference in order to support this affirmation.
- Table 8, please follow the same recommendation I gave before for Tables 5,6 and 7. If the table is going to be divided, you should put the head of the table in each part of the table
Author Response
Dear Reviewer,
Thank you for your detail comments.
-we devided Table 1 on table anf two figures;
-Line 92 and 103 (now 101 and 111, respectively) was redacted;
-Table 2 was transformed such as one table and two figures;
-we didn't transformed Table 3 because this compounds (guianolides) may show in this form using total chemical structure;
-Line 125 (now 143) - several information about sesquiterpene lactone was added;
-Figure 3 (now Figure 4) was corrected;
-Table 4 was transformed to Figure 8, Table 5, 6, 7 were corrected as Table 4, 5, 6;
-Table 8 (now Table 7) was not transform, this information is summarized from acticle [21];
-section Abbreviations was added;
-Line 267 (now 261) - this information ahve reference [17] (Line 258, now 252).

Reviewer 2 Report
General comments :
The Topic is interesting and fit the scope of the journal, however, I’m not recommending the review to be published in plant MDPI journal in the actual form, a deep revision is needed. More particularly, I find this manuscript very confusing and very unclear. authors often detailed commercial Kits of biological activities very technically so they have to focus only on the results, many concerns exist in the chemical structures. English grammar and language mistakes are very frequent. List of abbreviations is absent that make the manuscript unacceptable.
More specifically,
From a chemical point of view :
- Line 119, coumarin instead cumarin
- In Figure 3 ; Line 122, these are coumarins and not lignans, replace
- Give real conformation of the sugar in all structures of the text
- Structures of flavonoids in table 1, position 4 is a carbonyl, coumarins must also be redone and corrected, give real conformations of sugar …..
- Make a Section of synthesis or remove from the text the paragraph below:
‘’The ketone of tetrahydrozaluzanin C is formed under the action of boron trifluoride from dihydroestafiatin; estafiatone is formed from estafiatin. Two derivatives of 131 estafiatone are formed under the action of μ‐chlorperoxybenzoic acid in chloroform: epoxide C10‐C14 and δ–lactone which is formed as a result of oxidation by Baeyer – Villiger. Zaluzanin C is 133 formed from zaluzanin D under the action of Escherichia coli as a result of acetate group separation. 134 Such transformations of zaluzanin D lead to formation of 11,13‐dihydrozaluzanin C, zaluzanin C, 135 estafiatone, dihydroestafiatone. It’s possible to synthesize dihydroestafiatone and zaluzanin C from α‐santonine with a yield of 2.5% in 15 steps. The key step is solvolytic regroupment (11S)‐3, 137 4‐epoxy‐1‐(mezilohy)‐eudesmano‐12‐6‐lactone.’’
- Correct structure of acanthiolide, carbon 13 in place of O
- The elemanolide structure is also false C13 instead O and must be redone, numbers are upside down
- Line 163, taraxasterol instead tarasterol
- Change presentation of table 4 p.7, it is not clear, in fact, alpha amyrin and beta amyrine are from two diffrent skeletons of triterpene, while the first is oleanane, the other is from ursane and can not really summurized together in the table in a such manner,
- From lupeol derivatives, the lenght of nonds must be respected, replace CH3 for example with only lines, so these structures have to be redone
- Table 5, Idem replace CH3 with lines and respect the lengths of the chemical bonds, sterols figure is to be redone for more visibility and numbering
- In table 5, compound 10, is it a Δ7 sitosterol ?, remove the fragment
- What is the name of the 4 compounds (polyacetylenes), please add
- Where is the R5 in tocol structures, correct the figure as well, in tocols there are phytyl (tocopherols) and prenylated chains (tocotrienols) but not a CH3 as reported
- Also figure of TOCOPHEROLS STRUCTURE ARE TO BE REDONE, REMOVE α from α‐CH3 in the phytyl chain of tocols, R2 and R4 are the same in the 8 forms of tocols, only the other R1 and R3 are differents, so please redone figure adequatly
- Line 225, 1-amyno ??? amino
- Line 226, rephrase please
- The parapgraphs from line 235 to 250 does not give any originality for the studied species and must therefore be eliminated
- Line 545, please correct
- 552, Cr, creatinine ?: please give a separate section of abbreviations related to the whole text
- Line 282, rain ???
- Species name always italic, see and correct example for line 303
- Analgetic or analgesic ?, two times in the text, replace please
- Tocols instead tocopherols in the title (line 217)
- Paragraphs between lines 237- 250 to be shifted under section polyphenols
- Line 323 INHIBITION INSTEAD INBITION
- Line 401, xanthine oxidase instead please correct
- Line 405 ? REPHRASE AND CORRECT
- LINE 327 give reference
- From line 440 to 461, this belong to the section of antiproliferative and cytotoxic activities
- Line 514 The females of white mice were treated by 20 ? water herbal extracts during 3 weeks, what is the difference between 20 water extracts, please give some details….
- Table 5 the prenylated chain must be corrected, lack of index 2 IN (CH‐CH3)
In the pharmacology section :
- Line 232-233, rephrase please, it is badly described
- paragraph 329-334 : A test on NK activity does not reflect direct antiproliferative activity. This is an effect on the ability of NK immune cells involved in the destruction of infected cells or tumor cells and the production of cytokines as well. In any case, this does not reveal a direct effect on cell proliferation. After that authors may rename the paragraph (antitumor properties or activities for example).
- Line 338, replace Using instead used.
- Line 344 Replace Cisplatin not cysplatin
- Lines 350-354 : there is far too much technical details. the kit caspase 3 activity and the length of detection should be removed, we can say that the extract has a proapoptotic activity that is accompanied by the activation of caspase , please rephrase as English also seems very bad to me.
- Lines 359-363 : Still too technical and in addition no conclusion on the result : Blockade in sub-G1 of the cell cycle (characteristic DNA degradation linked to cytotoxicity ?), please rephrase
- line 372. Replace with Tested or evaluated but not researched. Kiselova Y evaluated 23 aqueous plant extracts for their antioxidant activity please again rephrase…..
376-377 : Not clear what is this list (references ??)
Line 432 : in ACE-inhibitor activity : it seems to me that this is a test related to the ability of a molecule to inhibit hypertension, it's not related to the antiproliferative activity. The cytotoxic effects are not measured by ACE inhibitory effect. Everything is mixed……please correct
- Line 452 HUMAN INSTEAD HIMAN
- Line 509 analgetic ?
- Rephrase lines from 498 to 504, this not clear
Author Response
Dear Reviewer,
Thank you for detail comments.
Chemical point:
-Line 119 (now 135) was redacted;
-Figure 3, Line 122 (now Figure 4, Line 134) was corrected;
-we didn't have the aim to give detail infromation about conformation of sugars;
-Table 1, in structure of flavonoid position 4 was corrected;
-Line 127 (now Line 147), this paragraph was decreased;
-structure of acanthiolide and elemanolide (Figure 7) were corrected;
-Table 4 (now Figure 8) was transformed in more detail form;
-Table 5 (now Table 4) was corrected. Delta-7-sitosterol was also corrected;
-names of polyacetylenes were added;
-Table 7 (now Table 6) was cleaned;
-Line 225 (now Line 234) and Line 226 (now Line 235) were corrected;
-Lines 235-250 were transformed as Lines 235-236 now place Lines 141-142; Lines 237-250 place in Lines 91-99 and 114-120 (in sections Phenols and Flavonoids);
-Line 545 (now Line 541) was not correced according information in reference [65], this date of authours;
-Line 552 (now Line 549) was explained; Section Abbreviations was added;
-Line 282 (now Line 276), Line 303 (now Line 297) were corrected;
-Lines 505 and 509 (now Lines 503 and 508, respectively) -Analgetic was transformed to Analgesic;
-Line 217 (now Line 227 anf title of Table 6) was redacted (Tocopherols - Tocols);
-Lines 323, 401, 405, 327 (now Lines 317, 422-423, 427, 321, respectively) were corrected;
-Lines 440-461 were replaced to section Antitumor activity (now Lines 367-388);
-Line 514 (now Line 512) was rephrased;
-Table 5 (now Table 4) - this compounds doesn't consist prenyl groups (-CH2-CH=C(CH3)2); this compounds have -CH2-CH=CH-CH(CH3)2; we showed chains (as R1) including duble bond in main chain and -CH(CH3)2 as radicals. So this chains show in this form;
Pharmacology section:
-Lines 232-233 (now Lines 240-243) was rephrased;
-Paragraph 329-334 (now Lines 322-328) now have title Antitumor activity;
-Lines 338, 344 (now Lines 332, 338) were redacted;
-Lines 350-354 (now Lines 344-346) were rephrased;
-Lines 359-363 - this information was added with Lines 360-364;
-Lines 372, 376-377 (now Lines 391, 396-398, respectively) were corrected and added;
-Line 432 (now Line 455) was rephrased;
Line 452, 509 (now Lines 379, 507, respectively) were transformed;
-Lines 498-504 (now Lines 495-502) were rephrased.

Reviewer 3 Report
The topics described is very interesting and should be published but after major revisions. A review is not merely a list of what is contained in the articles mentioned but should contain critical issues to address the readers toward still unexplored aspects.
In synthesis:
Basically the authors does not follow the characteristic review scheme: discussion, is practically non-existent and the methodology is not mentioned at all as it leaves little information at the end in the section "Author Contributions", very limited. A review is not merely a list of information. A critical discussion should be introduced in order to address the reader toward still unexplored aspect of research on the topic. I.e., there are no information on the number of articles handled, selection criteria, how many dropped and why, etc. A decision three would be useful. The authors should follows the guides for the preparation of a review which are reported from several part.
The botanical information are interesting, but the ethnobotanical component is weak (I wouldn't use the title "Importance in medicine", too vague), the list of chemical composition and biological activities is fine, but in both cases they are lists without the slightest discussion, without comparative data with similar species or additional data concerning the bioactivity of the main molecules.
I have concerns on the reliability of the studies, especially those in silico. The authors should consider the cited paper and discuss this point.
English can be improved.
Author Response
Dear Reviewer,
Thank you for detail comments.
In Author's guides in MDPI the structure of Review acticle have front and back matter, references. The main text may be prepared according authours style.
The aim of this paper was collected and analyzed dates about chemical content and pharmacology researches onle one species, Onopordum acanthium L.
So we didn't showed information about ethnobotanical component, geographical dates. There are many articles and guides about biological effects of different agents on growth of thisle, ecological anf biological studies. This information also interesting and it may be separate article.
There is not many studies about chemical contents and pharmacology activity. Authours used different methods, so this dates are difficultly comparatived. Also this information may be comparatived with dates about another species of genus Onopordum. We will do this work.
Articles of authors Sharifi N. et al. have detailed informations. They carried out as in silico method and in vitro and in vivo methods to confirm results. We analyzed information from artciles, specially concerned in sections Materials anf Methods, Results and Discussion. Also chemical structures were inspected using PubChem database.

Round 2
Reviewer 2 Report
The subject is interesting, a lot of data but the writing and the English especially are really to be reviewedby a specialist in Scientific English, more specific comments are below:
Table 1. Scutellarein 4’‐methyl ether, correct the R4 position ?
Line 113, eriodictiol ? correct
Line 66 : 3. Ethnomedical uses instead medical using
Line 79 : remove or rephrase ‘’ thanks to its epidermal……’’
Line 81 structures instead structure
Line 83 : in the roots
Line 93 : from the Asteracea family
Figure 5, structure of the aesculin and aesculetin must have same dimension
Hexan extract should be replaced with hexane in whole text and table 7 as well
Line 514 correct, remove full stop
Line 142, leave space O.acanthium
Figure 8 the structure b and d must be redone, the C20 must be clearly seen, you can remove the number 20 but the structures have to be correct
Line 629 name in italic
Line 523 text to be continued in the same line after reference 63
Rephrase line 358-359
in parapgraph 367 -388, authors mix antitumor with antioxidant and antiinflammatory and antibacterial as well ; this should be reorganized
line 390, data instead dates
line 414 to 417, authors reported activities of the glutathion enzymatic system, they report the IC50 of some extracts, what represent these inhibitions ? this is somewhat confusing, we can’t understand the role of CAT, GPx and GST in these inhibitions….must be detailed?
line 324 I don’t understand what was done in the YAC, is it a test on cytotoxicity or test linked to NK cells ?), it is not clear
Line 430 rephrase, on the presence of antibacterial effect ? not understand
Line 493, decreased inflammation not inflammatory
Line 512, white mice is not scientific term, maybe balb-C, others ? Check please…
Line 513, after 3 weeks, give the dose…
Line 541, values of control must be reported
Line 550, which clinical stage after treament? (I, II or III)
-Paragraph 550 to 555 must be shifted under toxicity section
Line 555 : There is no comparison with losartan alone?
565 : this data not dates..
Line 374 ; remove ‘’antifungal activity is low’’
Line 24 (key word) use italic for the scientific name.
Author Response
Dear Reviewer,
Thank You for detail comments.
Table 1 - scutellarein 4'-vethyl ether, R4 position was corrected (H instead =C(O));
Line 113 (now Line 119) was corrected;
Lines 66, 79, 81, 83, 93 (now Lines 66, 79, 81, 83, 96) were redacted;
Figure 5, dimensions of the aesculin and aesculetin were transformed;
"hexan" was corrected to "hexane";
Line 514 (now Line 572) - what did you mean?
Line 142 (now Line 149) was replaced;
Figure 8 - chemical structures were cleaned;
Line 629 (now Line 735) was corrected;
Line 523 (now Line 391) was replaced on section Antitumor activity;
Paragraph 367-388 was divided and replaced in section Antitumor, Anti-inflammatory anf Antibacterial activity (now Lines 401-435, 341, 509);
Line 390 (now Line 437) was corrected;
Lines 414-417 (now Lines 461-465) was detailed;
Line 324 (now Line 345) was more detailed;
Line 430 (now Line 478) was rephrased;
Lines 493, 512, 513, 541 (now Lines 551, 570, 571, 635) was corrected and detailed;
Line 550 - there is not information in [65] about clinical stage after treatment, only quantitative data of blood pressure (mmHg);
Paragraph 550-555 (now 581-585) was replaced in section Toxicity;
Line 555 - Losartan was treated before treatment with Onopordum acanthium dry extract in the dose 50 mg/day. There is data about base blood pressure.
Line 565 (now Line 654) was corrected;
Line 374 was deleted;
Line 24 was corrected.

Reviewer 3 Report
The topic of the review is interesting and deserve publication, but not in the present form.
A review should be a "critical" work and not a simple collection of papers content that can be achieved by the readers using the on-line abstracts.
The authors should critically comment the content of each article with reference to the reliability of the data contained in the papers in relation to the methods used to achieve results.
Because the topic is very interesting, I strongly encourage the authors to revise the manuscript better following the rules for a review: what is the search method used? What the DataBases consulted (they are cited in author contributions)? What are the Criteria? How many paper consulted? How many discharged and why? Why so old literature reported? A decision three would help the reader to understand.
In more detail, if pharmacological studies are reviewed, one must assess the quality of the studies under review. Often pharmacological activities are reported very naively, and such papers are especially common in journals with no or poor peer-review. This information must be assessed (as much as it is possible based on the published papers) in terms of their scientific quality and validity.
Furthermore, how is this activity linked to local and traditional uses? In the last years, the debate about the validity of (especially pharmacological) data has intensified, in these regards the part on traditional use (Importance in Medicine) needs to be expanded incorporating more sources (the authors claims a widespread use of the plant).
Conclusions are very poor, need to be critical and specific, highlighting the achievements and specific scientific gaps in our knowledge. What conceptual and methodological problems have you identified in the papers reviewed? What further specific research on this topic should have priority? Otherwise the paper is a merely list of informations.
Also the abstract needs to be more precise and provide a critical assessment
English language needs to be improved, some sentences have no correct meaning due probably to self correction of the software. As an example: "There are different dates about antioxidant activity". Dates = Days
In the present form the paper needs major improvements before publication.
Author Response
Dear Reviewer,
thank you for detail comments.
We analyzed DataBases such as PubMed, ScienceDirect, ResearchGate, GoogleScholar, Mendeley at key words.
Summary it was found about 600 references. From this data we selected 98 english references and 60 russian references about Onopordum acanthium including common sources (16 eng and 20 rus), anatomy studies (5 eng and 2 rus), phytochemistry and bioactive data (44 eng and 18 rus), using in industry (5 eng), ecological and biological studies (28 eng), patents (20 rus).
In this paper we present data about chemical content and bioactive studies. All references were detail analysed using sections Materials and Methods, Results, References. All data in articles were statistically processed. Metadata of references were found in Mendeley and Zotero.
Because there are not many articles about O. acanthium and different methods are used in this works we can't used any criteria for systematic review and meta-analyses using PRISMA guidelines.
We corrected article, added information about tradicional use, also corrected conclusions. Table 7 demostrate pharmacological activity.

Round 3
Reviewer 2 Report
- English still not good, authors must ask inevitably for a Native English Speaker before resubmitting the paper,
- Line 21 : sentences not complete, these properties and not this……
- Fig2 resize the structures idem for fig3
- Table 4 for steroid, line 5 , R1 formula is not correct
s
- Line 129, linans such as pinoresinol……..and medioresinol
- Line 156 : Elemanolides and Eudesmanolides
- Line 157, replace connected with asssociated
- Line 161, replace then was by followed by
- Line 163 ; remove : ‘’There are 2,2 mg of 3 and 0,9 mg of 4 from 251 g of air-dried leaves.’’
- Line 169, add the before leaves and remove ‘’of capitulas and cypselas ‘’
- Line 173, replace contented by contained
- Line 175, replace’’ in generative parts of plant’’ by plant parts
- Line 178: remove ‘’ receptacles – 2,8%, leaves of involucre – 0,13%, pappus of cypselas – 0,7%, roots – 0,1%.’’ And replace it by ‘’ among other botanical parts’’
- Line 179, are you sure only ‘’cypselas’’ or flowers?
- In figure 8, structures b and d, C20 still not visible,
- Line 184: cypselas or flowers??
- Line 280, this effect? Which one, this is not clear, authors must begin with, “” the anti-inflammatory action of (namae of plant or extracts) described below should be connected….
- Line 299, at the dose instead in the dose
- Line 306, expression of NFKB
- Pragraph between lines 310 and 317 (XTT test, ) must be under section antiproliferatives, cytotoxic or antitumor, it is more adequate…
Table 6, title must be modified, the anti-inflammatory activity of the main……
324: expression of COX
328, rephrase
332 rephrase
Line 344 IC50 instead EC50
350-351, remove description of the trypan blue assay, it is very known
356; HL-60 cells
358: cell cycle instead cell living cycle
386 rephrase….
388 In the dose IC50 must be replaced by ‘’with an IC50 of ‘’
401 these instead this
404 inhibition of COX must be under section antiinflammation and proton not protone
Etc etc…;
Author Response
Dear Reviewer,
Thank you for detail comments.
Line 21 - sentence was corrected, During all text we replaced "this" by "these".
Fig2 - size of these structures have similar ratio;
Table 4, steroid 5 - we changed this fragment according chemical structure at PubChem "Stigmasterol", but it's may be two forms of this radical depending on the main chain;
Line 129 (now Line 146) - corrected;
Line 156 (now Line 175) and Line 157 (now Line 176) - corrected;
Line 161 (now Line 180), Line 163 (now Line 182) - corrected;
Line 169 (now Line 188) - was added "the", but in these several articles authors describe experiments with capitulum and achenes and parts of these plant materials;
Line 173 (now Lines 192-193, 196) - corrected;
Line 175 (now Line 194) - "generative" was changed by "reproductive";
Line 178 (now Line 197) - in corresponding articles these parts of plant are described in similar maner;
Line 179 (now Line 198) - these compounds were found in fat oil of achenes;
Figure 8 - C20 is now visible;
Line 184 (now Lines 203-204) - during all text we replaced "cypsela" by "achenes";
Line 280 (now Line 299) - corrected;
Line 299 (now Line 309) - during all text we replaced "in the dose" by "at the dose"
Line 306 (now Line 317) - during all text we added "of" in similar sentences;
Lines 310-317 (now Lines 355-361) were replaced;
Table 6, title was corrected;
Line 344 (now Line 354) - authors in corrresponding article use EC50;
Line 350-351 (now Line 365) - removed;
Line 356 (now Line 370) - corrected;
Line 358 (now Line 372) - corrected;
Line 386 (now Line 398) - rephrased;
Line 388 (now Line 400) - corrected;
Line 404 (now Lines 392-396) - replaced.

Reviewer 3 Report
The review is still weak: as an example compare Germacranolides and triterpene alcohols. The second one is correct, several data are cited useful to the readers. All components and chapter should be developed in the same manner giving more data taken from original article consulted by the authors.
The authors describes the strategy followed for doing the review in the answer to the referee. This should be reported in the manuscript at the beginning of the article.
Finally English is still poor, I have highlighted a few meaningless phrases just to give an example.
After this modification the paper can be accepted for pubblication
Author Response
Dear Reviewer,
Thank you for comments.
We added chapter "Methodology of review".
Also we corrected data about chemical content, but in this chapter we demonstrate the main chemical groups and describe quantitative content according corresponding articles.
